# Cariprazine in an Adolescent with Tourette Syndrome with Comorbid Attention Deficit Hyperactive Disorder and Depression: A Case Report

**DOI:** 10.3390/healthcare11182531

**Published:** 2023-09-13

**Authors:** Jesjeet Singh Gill

**Affiliations:** Department of Psychological Medicine, University Malaya, Kuala Lumpur 50603, Malaysia; jesjeet@um.edu.my

**Keywords:** cariprazine, Tourette syndrome, major depressive disorder, ADHD, tics

## Abstract

Tourette syndrome is a complex neuropsychiatric condition that manifests in childhood and is often associated with other psychiatric comorbidities. This case report describes a young male with Tourette syndrome with major depressive disorder and attention deficit hyperactivity disorder (ADHD) who experienced troublesome side effects due to his existing medications (escitalopram, risperidone, and methylphenidate). In order to control his tics, ameliorate depressive symptoms, and eliminate side effects of stiffness and sedation, risperidone was switched to cariprazine, a third-generation antipsychotic medication with D3-D2 partial agonism. In addition, the antidepressant dose was also increased. With the new combination, the patient reported good control of his tics, together with significant improvement in depressive symptoms and no side effects. Based on this case and the reviewed literature, cariprazine might be a viable option for patients with Tourette syndrome with other comorbid illnesses who are prone to side effects of medication.

## 1. Introduction

Tourette syndrome is a complex neuropsychiatric condition characterized by the presence of tics, i.e., sudden, repetitive, and stereotyped motor movements that affect discrete muscle groups. It is a unique neuropsychiatric condition as it sits at the interface of neurology (a movement disorder) and psychiatry (a behavioral disorder) [1]. The latest Edition of the Diagnostic and Statistical Manual of Mental Disorders (DSM 5) states that the presentation should consist of not only motor tics but the occurrence of vocal tics as well, but not necessarily at the same time. The tics may wax and wane but must persist at least for 1 year after the onset. It also states that the onset must occur before the age of 18 years old [2]. It is estimated to affect about 1% of children and adolescents with a typical age of onset between the years of 4 to 6, usually following a stressful life event, such as the start of school, being bullied, or any psychosocial stressors [3,4]. The most severe symptoms are usually apparent between the ages of 10 to 12 years [4]. In most cases, a marked reduction in the severity and frequency of tics is expected over time regardless of medical treatment, but in some cases the symptoms may persist into adulthood [5]. The prevalence of Tourette syndrome in adults is low, with only about 0.05% being affected [6], and it is more common in males than females [3].

Tourette syndrome often coexists with other psychiatric conditions, such as comorbid attention deficit hyperactivity disorder (ADHD), obsessive compulsive disorder, and affective disorders including major depressive disorder and anxiety disorders [2,7]. Indeed, affective symptoms can be present in up to 76% of Tourette syndrome patients, with around 10% meeting the diagnostic criteria for major depressive disorder [8]. A study by Chou et al. even found that the risk of developing major depressive disorder is five times more likely in those afflicted with Tourette syndrome compared to the general population [9]. In terms of attention deficit hyperactivity disorder, more than 50% of patients with Tourette syndrome can be affected with it [10]. Males with Tourette syndrome are more likely to have a comorbid attention deficit hyperactivity disorder compared to females and tend to have a poorer outcome. This many comorbidities is associated with greater functional, social, and academic impairment, impacting the quality of life of patients negatively [11]. Tourette syndrome adversely affects a child’s self-esteem and relationships with family and friends. This is especially worse during episodes when motor tics wax, accompanied with vocal tics that may persist non-stop for several hours [3].

The etiology of Tourette syndrome is most likely multifactorial, with many causes being studied and proposed. Family and twin genetic studies have provided strong evidence that genetic factors play a large role in whether a person ultimately develops Tourette syndrome. Other etiological causes that have been proposed include perinatal trauma, severe psychosocial trauma, and drug abuse, particularly stimulants [3].

The treatment of Tourette syndrome is complex. It is made difficult by the variability of different presentations that can occur among different individuals. Various studies have yet to find one single medication that can be effective in every person with Tourette syndrome. To conclude whether a particular medication is efficacious in a particular child is made difficult by the fact that tics can come and go for prolonged periods at a time. A medication that appears to work in the beginning may turn out to be non-efficacious when the symptoms wax. A trial-and-error strategy is likely needed to determine the best medication, or treatment combinations in those with more complex symptoms, such as those with concurrent obsessive compulsive, attention deficit hyperactive, or depressive symptoms. In these types of situations, especially involving children, there is a real concern regarding the use of these medications at a young age, together with the potential of drug interactions [12]. 

Tics are usually treated with dopamine blockers, i.e., antipsychotics, given the notion that greater reduction in tics (80–90%) can be achieved with drugs that confer greater dopamine antagonism [12]. Current antipsychotic medications that are indicated for the treatment of Tourette syndrome include pimozide, haloperidol, and aripiprazole [12,13,14]. Studies investigating the efficacy of risperidone have reported positive outcomes; therefore, risperidone is also frequently prescribed for Tourette syndrome [12,15]. However, antipsychotics are usually associated with numerous troubling side effects. Risperidone can cause weight gain, sedation, extrapyramidal symptoms, and prolactin elevation. Aripiprazole has been known to cause drowsiness, akathisia, restlessness, and sleep disturbance [16]. For cases with comorbid depression and anxiety, serotonin reuptake inhibitors (SSRIs) are often utilized, while stimulants such as methylphenidate are used for when symptoms of attention deficit hyperactivity disorder are present [11,17]. 

Besides antipsychotics, other drugs that have been used include drugs such as benzodiazepines, clonidine, and nicotine. Behavioral techniques such as habit reversal training and exposure response prevention have been found to play a role in treating Tourette syndrome, and they may be used alone or in conjunction with pharmacological options. For the more severe resistant cases, surgical interventions such as deep brain stimulation (DBS) may play a role. Electroconvulsive therapy (ECT) and repetitive Transcranial magnetic stimulation (rTMS) have also been utilized [12]. 

Cariprazine is a dopamine D_3_, D_2_, and 5-HT_1A_ partial agonist and antagonist at the 5-HT_2B_ receptors with a higher affinity to D3 receptors [18]. It is categorized as a third-generation antipsychotic medication and is currently approved by the USFDA for the treatment of adult schizophrenia [19,20,21,22] and bipolar I disorder patients (manic, mixed, and depressed episodes) [23], as well as for the adjunctive treatment of major depressive disorder [24,25]. Given its similar receptor profile to aripiprazole but with increased affinity for D_3_ receptors, it might be a good treatment option for the treatment of Tourette syndrome with comorbid depression. 

The aim of the present case report is to show the effectiveness of the novel antipsychotic medication, cariprazine, in reducing tics, ameliorating depressive symptoms, and maintaining a reduction in tics without producing debilitating side effects in a young male patient [12].

## 2. Case Presentation

An 18-year-old male student of Asian Indian ethnicity presented to us with prior diagnoses of Tourette syndrome, major depressive disorder, and ADHD while still experiencing low mood, impaired attention, tics, as well as side effects (sedation and stiffness) due to his previously prescribed medication. 

The first signs of illness started when the patient was 13 years old when he made the transition from primary to secondary school. He found it immensely difficult to focus on schoolwork which made him feel stressed and eventually led to the development of motor tics. These initially manifested as intermittent puckering of his lips. Occasionally, he would also experience sudden neck jerky movements. Later, the patient started to experience vocal tics as well which manifested in sudden grunting sounds. This had caused him much embarrassment and led him to have low self-esteem as well. His classmates would frequently make fun of him and, at times, avoid him all together. Despite all this, he could still cope with his studies. However, at the age of 17 years old, he started to experience a low mood and anxiety as well. This was associated with poor concentration, loss of interest, excessive eating, increased appetite, and lethargy. This was when his studies started to deteriorate, and he started failing several exam papers. He felt guilty that he had let his mother down and had experienced some passive death wishes without any active plans. 

Personal history revealed an adverse childhood period, where his father passed away suddenly when he was an infant; hence, he grew up without a father figure. His mother was also seldom at home as she had to take up several jobs to make ends meet. He was often taken care of by his elder sister, who was 5 years older than him. He also has a strong family history of psychiatric illness where both his elder sister and mother were diagnosed with major depressive disorder. His mother also had several admissions to the psychiatric ward when he was younger. His maternal grandmother had committed suicide before he was born, but not much is known about the circumstances behind that. 

As his mother was worried about his progressively deteriorating grades, she brought him to consult a private physician. He was then diagnosed with major depressive disorder, ADHD, and underlying Tourette syndrome. To alleviate his symptoms, he was prescribed escitalopram (5 mg daily) to treat his depression, risperidone (0.5 mg daily) for his Tourette syndrome, and methylphenidate (5 mg daily) for his attention deficit hyperactivity disorder. Over the next several weeks, escitalopram was titrated up to 10 mg daily and risperidone was titrated up to 2 mg daily, while methylphenidate was maintained at 5 mg daily. After six months, the patient was referred to us for continuation of care as he could not afford the charges at the private physician’s clinic.

During the first visit, the patient reported that initially his tics were fairly well controlled, but he experienced sedation and stiffness with risperidone. Thus, he stopped the risperidone a month prior, causing his tics to worsen. Although, he was adherent to methylphenidate and escitalopram. He did report some improvement in his attention and concentration with methylphenidate, but this was not completely satisfactory. He also reported that his depressive symptoms were only about 60% better (according to his own evaluation). Though his mood had somewhat improved, he still had hypersomnia, lethargy, and poor concentration on his studies. We decided not to restart the risperidone due to the bothersome side effects it had caused him before. Methylphenidate was maintained at a dose of 5 mg daily, but we decided to increase the dose of escitalopram to 15 mg a day as he was still having significant depressive symptoms. Cariprazine was initiated at a dose of 1.5 mg a day, with the dual aim of controlling his tics due to his Tourette syndrome as well as to augment the effect of the antidepressant. 

Cariprazine was specifically chosen for several reasons. First of all, the patient responded to risperidone (D_2_ antagonist) previously; hence, cariprazine’s partial agonist activity at the dopamine receptors could confer the same benefit but with a decreased likelihood of side effects. Secondly, given the recent evidence regarding cariprazine’s effectiveness as an adjunctive agent in depression, it could also provide additional help in ameliorating the patient’s depression with its partial agonism at the 5-HT_1A_ receptors. Finally, cariprazine is also an antagonist at the serotonin 5-HT_2B_ receptor, which is considered to have an adjunctive effect in the presence of selective serotonin reuptake inhibitors. 

Throughout the subsequent visits, the dosages of all concomitant medication remained unchanged from his first consultation with us. In just 2 weeks after his first consultation with us, he reported improvement in his mood, attention, and concentration, and subsequently improvement in his studies. Hence, we decided to keep the dosage of escitalopram the same. He continued to report improvement in his mood in subsequent visits, and after around 8 months from his initial consultation with us, he felt that he had returned back to his previous euthymic state. 

As for the occurrence of his tics, he began to notice a reduction in their frequency after about 3 weeks on cariprazine. After about 2 months, he did not experience any more vocal tics, and motor tics completely ceased after 4 months. Hence, we did not see the need to increase the dose of cariprazine. Similarly, his dose of methylphenidate remained the same right from the first consultation as he showed improvement in all his symptoms early on. 

He remained well throughout subsequent visits. More importantly, he did not complain about being sedated or stiff anymore, and no additional side effects have emerged while he has been on this new medication regime for the last 14 months. Though non-pharmacological approaches such as behavior therapy can be utilized in Tourette syndrome, we did not try it in this case as he responded early to pharmacological treatment. 

As this case was treated in a non-research setting, structured scales were not used to monitor his progress. The improvements in this patient’s symptoms were gauged subjectively throughout his follow ups. In retrospect, using structured scales such as the Yale Global Tic Severity Scale (YGTSS) [26] would have been more accurate to measure improvement.

## 3. Discussion

To the best of our knowledge, this is the first case that describes the efficacy of cariprazine in ameliorating the symptoms of Tourette syndrome in an adolescent with comorbid major depressive disorder and ADHD. Although there is currently no evidence regarding the efficacy of cariprazine in ameliorating tics, aripiprazole, which is also a dopamine partial agonist, has been shown to reduce tics with similar effect sizes as haloperidol and risperidone [26], providing a solid basis for why cariprazine could also work in Tourette syndrome with comorbid major depressive disorder and attention deficit hyperactivity disorder. 

The efficacy of cariprazine in depressive symptoms was established in several Phase I clinical trials in patients with bipolar I disorder in the depressive phase [27,28]. In an 8-week randomized, double-blind, placebo-controlled study in adults with bipolar depression, cariprazine at a dose of both 1.5 mg/day and 3 mg/day showed significantly greater improvement in the Montgomery–Åsberg Depression Rating Scale change in the total score from the baseline to week 6 compared with placebo [27]. Another was a double-blind placebo-controlled Phase III trial by Early et al. Cariprazine, at doses of either 1.5 mg/daily and 3 mg/daily, was superior in reducing depressive symptoms in subjects with bipolar I disorder experiencing a depressive episode [28].

Cariprazine has also been shown to be efficacious when used as an adjunct in patients with major depression with incomplete response to standard antidepressants, as shown in two studies. In the first study, by Durgam et al., the reduction in the Montgomery–Åsberg Depression Rating Scale total score at week 8 was significantly greater with adjunctive cariprazine at doses between 2 mg and 4.5 mg daily compared to placebo [24]. The second study was a 19-week placebo-controlled Phase II study that demonstrated cariprazine at doses of 1–2 mg a day; it showed greater improvement in the Montgomery–Åsberg Depression Rating Scale when used as an adjunctive agent to antidepressants in subjects with treatment-resistant depression [25]. 

Cariprazine’s unique receptor profile, with regard to both dopamine and serotonin receptors, is thought to be the reason why it is efficacious in many different symptom domains. It is a partial agonist at both the D_3_ and D_2_ dopamine receptors, but its affinity towards D_3_ receptors is 10-fold higher than towards D_2_ receptors [29]. Its affinity towards D_3_ receptors is even higher than intrinsic dopamine, causing it to act like a dopamine antagonist. This D_3_ antagonism at the ventral tegmental area of the brain leads to dopamine release at the prefrontal cortex, in turn stimulating the D_1_ receptors. This is postulated to improve cognition and mood, as well as treat negative symptoms of schizophrenia. Cariprazine’s D_2_ partial agonism at the mesolimbic pathway is believed to treat the positive symptoms of schizophrenia [18]. 

With regard to the serotonin receptors, it has a high affinity towards the 5-HT_2B_ receptors, where it acts as an antagonist. Additionally, it acts as a partial agonist at the 5HT_1A_ receptors, with a moderate, but still significant, affinity. This action on 5HT_1A_ is thought to produce an antidepressant-like effect [29]. 

This positive effect on symptoms of major depressive disorder was seen in this case, when the patient received cariprazine 1.5 mg/day together with escitalopram 15 mg/day. Compared to the previously achieved 60% improvement in mood symptoms with the combination of risperidone and escitalopram, the patient reported greater improvement in depressive symptoms when he switched to this regime. In fact, after just slightly over 3 months on this combination of medication, he declared himself as having recovered from his depression. Being able to continue schoolwork, he did not mention any problems with his attention, a cognitive domain that might have also improved due to the D3 activity of cariprazine [30].

When he was on the previous medication regime, the patient complained of being sedated and stiff, which are frequently described side effects of antipsychotic medications such as risperidone [31]. In contrast, cariprazine is considered an activating compound, with insomnia being a more likely side effect than sedation or somnolence [32]. This activating effect may also contribute to the reduction in depressive symptoms. Being a dopamine partial agonist, cariprazine is unlikely to produce extrapyramidal side symptoms such as stiffness, which was experienced by the patient when he was on risperidone. Lacking anti-histaminergic actions also lessens the likelihood of causing sedation and weight gain [33]. In this patient, no side effects were reported after the switch to cariprazine, which is important in treating young patients who are still studying. 

As for now, the safety of cariprazine in adolescents is yet to be established in large clinical trials. However, in a review by Tohen, cariprazine was found to be safe in adolescents aged between 13 and 18 years old. In a 4-week trial involving 43 adolescents, no fatalities or serious adverse events were recorded. All adverse events were reported as being either mild or moderate, with only one incident (headache and tonsillitis) reported as a severe event [34].

## 4. Conclusions

In conclusion, we believe that cariprazine can be an important choice when choosing a medication to treat Tourette syndrome, particularly among those who are suffering from comorbid conditions such as major depressive disorder or ADHD and are prone to the common side effects of antipsychotics. However, as this is a single case report, more research is needed to determine if cariprazine can indeed be an effective option in Tourette syndrome, as well as being safe and tolerable across all age groups. This case also highlights that personalized treatment is usually the norm, rather than the exception, when treating complex neuropsychological conditions such as Tourette syndrome that frequently present with comorbidities.

## Data Availability

Research data are available upon reasonable request to the corresponding authors.

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
