# Peer review of "Cariprazine in an Adolescent with Tourette Syndrome with Comorbid Attention Deficit Hyperactive Disorder and Depression: A Case Report"

_healthcare, 2023, doi:10.3390/healthcare11182531_

Round 1
Reviewer 1 Report
Comments and Suggestions for Authors: The present work presents insights on the use of cariprazine for the treatment of Tourette syndrome, in a patient with comorbid conditions such as Major Depressive Disorder or Attention Deficit Hyperactivity Disorder. Abstract está claro e objetivo ao trabalho e resultados apresentados. Some clarifications are needed: - It is important to say how long it took to notice improvement in symptoms after using cariprazine. - A broader approach to the safety profile and use in adolescents is needed. For example, this paper presents an overview of cariprazine use in various age groups, including children and adolescents. (Tohen M. Cariprazine as a Treatment Option for Depressive Episodes Associated with Bipolar 1 Disorder in Adults: An Evidence-Based Review of Recent Data. Drug Des Devel Ther. 2021 May 12;15:2005-2012. doi: 10.2147/DDDT.S240860. PMID: 34012253; PMCID: PMC8126799.). - In the discussion of the results, it can be expanded to better clarify the level of receptors (D3-D2 , 5HT1A ,5-HT2B....).- The text showed a high level of similarity with other works, I suggest checking this issue using an anti-plagiarism program.
Author Response
Thank you for your kind comments and useful suggestions. My replies to your queries are as follows.
#1. How long before improvement was seen with cariprazine.
It was 3 weeks. I have also added the following to the text, to elaborate more on the improvement in all symptoms:
“Throughout the subsequent visits, the dosages of all concomitant medication remained unchanged from his first consultation with us. In just 2 weeks after his first consultation with us, he reported improvement in his mood, attention and concentration, and subsequently improvement in his studies. Hence we decided to keep the dosage of escitalopram the same. He continued to report improvement in his mood in subsequent visits, and after around 8 months from his initial consultation with us, he felt that he had returned back to his previous euthymic state.
As for the occurrence of his tics, he began to notice a reduction in frequency after about 3 weeks on cariprazine. After about 2 months, he did not experience any more vocal tics, and motor tics completely ceased after 4 months. Hence we did not see the need to increase the dose of cariprazine. Similarly, his dose of methylphenidate remained the same right from the first consultation as he showed improvement in all his symptoms from early on.”
#2. Regarding safety profile in adolescents, I’ve added the following to the text
“As for now, the safety of cariprazine in adolescents is yet to be established in large clinical trials. However, in a review by Tohen et al, cariprazine was found to be safe in adolescents aged between 13 to 18 years old. In a 4 week trial involving 43 adolescents, no deaths or serious adverse events were reported. All treatment emergent adverse events were reported as mild to moderate, with only 1 incident (headache and tonsillitis) reported as a severe event”.
#3. I’ve added more to the discussion about the level of receptors in the text, as follows:
“Cariprazine’s unique receptor profile with regards to both dopamine and serotonin receptors is thought to be the reason why it is efficacious in many different symptom domains. It is a partial agonist at both the D2 and D3 dopamine receptors, but it’s affinity towards the D3 receptors is 10 fold higher than towards D2 receptors [29]. It’s affinity towards the D3 receptors is even higher than intrinsic dopamine causing it to act like a dopamine antagonist. This D3 antagonism at the ventral tegmental area of the brain leads to dopamine release at the prefrontal cortex that in turn stimulates the D1 receptors. This is postulated to improve cognition and mood, as well as treat negative symptoms of schizophrenia. Cariprazine’s D2 partial agonism at the mesolimbic pathway is believed to treat the positive symptoms of schizophrenia [18].
With regards to the serotonin receptors, it has high affinity towards the 5-HT2B receptors where it acts as an antagonist. Additionally, it acts as a partial agonist at the 5HT1A receptors, with a moderate, but still significant affinity. This action on 5HT1A is thought to produce an antidepressant-like effect [29].”
#4. After revising the script, I checked the text (excluding references) on an anti-plagiarism tool (Turnitin) and found a similarity level of only 9%.
Reviewer 2 Report
This case report presents an interesting and unique case of a young male with Tourette syndrome compounded by depression and ADHD. The authors explored the effectiveness of cariprazine for controlling tics, managing depressive symptoms, and minimizing side effects. The case report provides valuable insights into potential treatment strategies for patients with Tourette syndrome and comorbid conditions. The manuscript is well-written and organized, with a comprehensive literature review, a clear case presentation, and discussions on the implications. I have the following comments/questions for the author to enhance the clarity and depth of this report.
#1. Case presentation, lines 163-168 - I was curious about the trajectory of dose adjustments and maintenance dose of cariprazine as well as any changes to other concomitant medications over the 14-month follow-up period. Please elaborate on them if applicable.
#2. Has this patient ever pursued non-pharmacological treatment approaches?
#3. To strengthen the conclusion, it could be helpful to briefly discuss the need for further research to establish the broader applicability of cariprazine in similar cases and emphasize the importance of personalized treatment approaches for complex neuropsychiatric conditions.
Author Response
Thank you for your kind comments and useful suggestions. My replies to your queries are as follows.
#1. I have elaborated about the dosages of medication over the follow up period and have added the following in the text:
“Throughout the subsequent visits, the dosages of all concomitant medication remained unchanged from his first consultation with us. In just 2 weeks after his first consultation with us, he reported improvement in his mood, attention and concentration, and subsequently improvement in his studies. Hence we decided to keep the dosage of escitalopram the same. He continued to report improvement in his mood in subsequent visits, and after around 8 months from his initial consultation with us, he felt that he had returned back to his previous euthymic state.
As for the occurrence of his tics, he began to notice a reduction in frequency after about 3 weeks on cariprazine. After about 2 months, he did not experience any more vocal tics, and motor tics completely ceased after 4 months. Hence we did not see the need to increase the dose of cariprazine. Similarly, his dose of methylphenidate remained the same right from the first consultation as he showed improvement in all his symptoms from early on”.
#2. Regarding non pharmacological approaches, I've added this to the text:
“Though non-pharmacological approaches such as behavior therapy can be utilized in Tourette syndrome, we did not try it in this case as he responded early to pharmacological treatment.”
#3. I have added this after your suggestion:
“However, as this is a single case report, more research is needed to determine if cariprazine can indeed be an effective option in Tourette Syndrome, as well as being safe and tolerable across all age groups. This case also highlights that personalized treatment is usually the norm, rather than the exception when treating complex neuropsychological conditions, as Tourette Syndrome frequently presents with comorbidities.”
Thank You very much
Reviewer 3 Report
Overall well written case report manuscript. This is a case report that uses a newer antipsychotic, Cariprazine off label in an adolescent with Tourette Syndrome with Comorbid ADHD and Depression.
I feel authors should consider using a rating scale like Yale Global Tic Severity Scale (YGTSS) to demonstrate efficacy of Cariprazine.
https://www.ncbi.nlm.nih.gov/pmc/articles/PMC7949908/
Otherwise, no identified major areas of weakness. Author has appropriately expanded on all sections of a case report that is comprehensive, as well as clinically relevant information that would be useful to the reader. No obvious flaws or weaknesses.
There is no previous published literature describing efficacy or experience with Cariprazine in an adolescent with Tourette’s syndrome. So this case report does add to literature.
Another suggestion, In line 106, specify Asian Indian, as the worldwide audience could confuse with Native Americans in North America, who are also known as Indians.
Author Response
Thank you for your kind comments and useful suggestions. My replies to your queries are as follows.
#1. Unfortunately, we did not use any scales to measure improvement, but would certainly take up your suggestion in the future. I also added the following to the manuscript text:
“As this case was treated in a non-research setting, structured scales were not used to monitor his progress. The improvements in this patient's symptoms were gauged subjectively throughout his follow ups. In retrospect, using structured scales such as the Yale Global Tic Severity Scale (YGTSS) would have been more accurate in measuring improvement.”
#2. I added the phrase “Asian Indian” in the text, as you correctly pointed out.
Thank you very much